# Co-regularized Alignment for Unsupervised Domain Adaptation

**Abhishek Kumar**
MIT-IBM Watson AI Lab, IBM Research
abhishk@us.ibm.com

**Prasanna Sattigeri**
MIT-IBM Watson AI Lab, IBM Research
psattig@us.ibm.com

**Kahini Wadhawan**
MIT-IBM Watson AI Lab, IBM Research
kahini.wadhawan@ibm.com

**Leonid Karlinsky**
MIT-IBM Watson AI Lab, IBM Research
leonidka@il.ibm.com

**Rogerio Feris**
MIT-IBM Watson AI Lab, IBM Research
rsferis@us.ibm.com

**William T. Freeman**
MIT
billf@mit.edu

**Gregory Wornell**
MIT
gww@mit.edu

## Abstract

Deep neural networks, trained with large amount of labeled data, can fail to generalize well when tested with examples from a *target domain* whose distribution differs from the training data distribution, referred as the *source domain*. It can be expensive or even infeasible to obtain required amount of labeled data in all possible domains. Unsupervised domain adaptation sets out to address this problem, aiming to learn a good predictive model for the target domain using labeled examples from the source domain but only unlabeled examples from the target domain. Domain alignment approaches this problem by matching the source and target feature distributions, and has been used as a key component in many state-of-the-art domain adaptation methods. However, matching the marginal feature distributions does not guarantee that the corresponding class conditional distributions will be aligned across the two domains. We propose co-regularized domain alignment for unsupervised domain adaptation, which constructs multiple diverse feature spaces and aligns source and target distributions in each of them individually, while encouraging that alignments agree with each other with regard to the class predictions on the unlabeled target examples. The proposed method is generic and can be used to improve any domain adaptation method which uses domain alignment. We instantiate it in the context of a recent state-of-the-art method and observe that it provides significant performance improvements on several domain adaptation benchmarks.

## 1   Introduction

Deep learning has shown impressive performance improvements on a wide variety of tasks. These remarkable gains often rely on the access to large amount of labeled examples $(x, y)$ for the concepts of interest ($y \in Y$). However, a predictive model trained on certain distribution of data ($\{(x, y) : x \sim P_s(x)\}$, referred as the *source domain*) can fail to generalize when faced with observations pertaining to same concepts but from a different distribution ($x \sim P_t(x)$, referred as the *target domain*). This problem of mismatch in training and test data distributions is commonly referred as domain or covariate shift [34]. The goal in *domain adaptation* is to address this mismatch and obtain a model that generalizes well on the target domain with limited or no labeled examples from

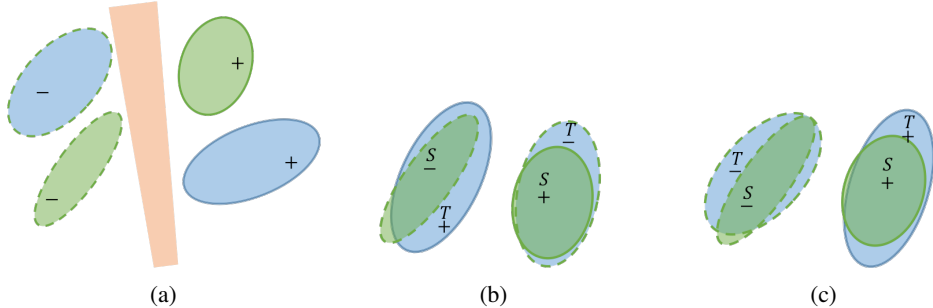

Figure 1: Example scenarios for domain alignment between source $S$ (green) and target $T$ (blue). Continuous boundary denotes the $+$ class and the dashed boundary denotes the $-$ class. (a) $P_s$ and $P_t$ are not aligned but $d_{\mathcal{H}\Delta\mathcal{H}}(P_s, P_t)$ is zero for $\mathcal{H}$ (a hypothesis class of linear separators) given by the shaded orange region, (b) Marginal distributions $P_s$ and $P_t$ are aligned reasonably well but expected error $\lambda$ is high, (c) Marginal distributions $P_s$ and $P_t$ are aligned reasonably well and expected error $\lambda$ is small.

the target domain. Domain adaptation finds applications in many practical scenarios, including the special case when source domain consists of simulated or synthetic data (for which labels are readily available from the simulator) and target domain consists of real world observations [43, 39, 5].

We consider the problem of *unsupervised domain adaptation* where the learner has access to only unlabeled examples from the target domain. The goal is to learn a good predictive model for the target domain using labeled source examples and unlabeled target examples. *Domain alignment* [13, 15] approaches this problem by extracting features that are invariant to the domain but preserve the discriminative information required for prediction. Domain alignment has been used as a crucial ingredient in numerous existing domain adaptation methods [17, 26, 41, 40, 4, 16, 44, 42, 35]. The core idea is to align distributions of points (in the feature space) belonging to same concept class across the two domains (i.e., aligning $g\#P_s(\cdot|y)$ and $g\#P_t(\cdot|y)$ where $g$ is a measurable feature generator mapping and $g\#P$ denotes the push-forward of a distribution $P$), and the prediction performance in target domain directly depends on the correctness of this alignment. However, the right alignment of class conditional distributions can be challenging to achieve without access to any labels in the target domain. Indeed, there is still significant gap between the performance of unsupervised domain adapted classifiers with existing methods and fully-supervised target classifier, especially when the discrepancy between the source and target domains is high[1].

In this work, we propose an approach to improve the alignment of class conditional feature distributions of source and target domains for unsupervised domain adaptation. Our approach works by constructing two (or possibly more) diverse feature embeddings for the source domain examples and aligning the target domain feature distribution to each of them individually. We co-regularize the multiple alignments by making them agree with each other with regard to the class prediction, which helps in reducing the search space of possible alignments while still keeping the correct set of alignments under consideration. The proposed method is generic and can be used to improve any domain adaptation method that uses domain alignment as an ingredient. We evaluate our approach on commonly used benchmark domain adaptation tasks such as digit recognition (MNIST, MNIST-M, SVHN, Synthetic Digits) and object recognition (CIFAR-10, STL), and observe significant improvement over state-of-the-art performance on these.

## 2   Formulation

We first provide a brief background on domain alignment while highlighting the challenges involved while using it for unsupervised domain adaptation.

## 2.1 Domain Alignment

The idea of aligning source and target distributions for domain adaptation can be motivated from the following result by Ben-David et al. [2]:

**Theorem 1 ([2])** *Let $\mathcal{H}$ be the common hypothesis class for source and target. The expected error for the target domain is upper bounded as*

$$\epsilon_t(h) \leq \epsilon_s(h) + \frac{1}{2} d_{\mathcal{H}\Delta\mathcal{H}}(P_s, P_t) + \lambda, \forall h \in \mathcal{H}, \tag{1}$$

*where $d_{\mathcal{H}\Delta\mathcal{H}}(P_s, P_t) = 2 \sup_{h,h' \in \mathcal{H}} |\mathrm{Pr}_{x \sim P_s}[h(x) \neq h'(x)] - \mathrm{Pr}_{x \sim P_t}[h(x) \neq h'(x)]|$, $\lambda = \min_h[\epsilon_s(h) + \epsilon_t(h)]$, and $\epsilon_s(h)$ is the expected error of $h$ on the source domain.*

Let $g_s : X \to \mathbb{R}^m$ and $g_t : X \to \mathbb{R}^m$ be the feature generators for source and target examples, respectively. We assume $g_s = g_t = g$ for simplicity but the following discussion also holds for different $g_s$ and $g_t$. Let $g\#P_s$ be the push-forward distribution of source distribution $P_s$ induced by $g$ (similarly for $g\#P_t$). Let $\mathcal{H}$ be a class of hypotheses defined over the feature space $\{g(x) : x \sim P_s\} \cup \{g(x) : x \sim P_t\}$ It should be noted that alignment of distributions $g\#P_s$ and $g\#P_t$ is not a necessary condition for $d_{\mathcal{H}\Delta\mathcal{H}}$ to vanish and there may exist sets of $P_s, P_t$, and $\mathcal{H}$ for which $d_{\mathcal{H}\Delta\mathcal{H}}$ is zero without $g\#P_s$ and $g\#P_t$ being well aligned (Fig. 1a). However, for unaligned $g\#P_s$ and $g\#P_t$, it is difficult to choose the appropriate hypothesis class $\mathcal{H}$ with small $d_{\mathcal{H}\Delta\mathcal{H}}$ and small $\lambda$ without access to labeled target data.

On the other hand, if source feature distribution $g\#P_s$ and target feature distribution $g\#P_t$ are aligned well, it is easy to see that the $\mathcal{H}\Delta\mathcal{H}$-distance will vanish for any space $\mathcal{H}$ of sufficiently smooth hypotheses. A small $\mathcal{H}\Delta\mathcal{H}$-distance alone does not guarantee small expected error on the target domain (Fig. 1b): it is also required to have source and target feature distributions such that there exists a hypothesis $h^* \in \mathcal{H}$ with low expected error $\lambda$ on both source and target domains. For well aligned marginal feature distributions, having a low $\lambda$ requires that the corresponding class conditional distributions $g\#P_s(\cdot|y)$ and $g\#P_t(\cdot|y)$ should be aligned for all $y \in Y$ (Fig. 1c). However, directly pursuing the alignment of the class conditional distributions is not possible as we do not have access to target labels in unsupervised domain adaptation. Hence most unsupervised domain adaptation methods optimize for alignment of marginal distributions $g\#P_s$ and $g\#P_t$, hoping that the corresponding class conditional distributions will get aligned as a result.

There is a large body of work on distribution alignment which becomes readily applicable here. The goal is to find a feature generator $g$ (or a pair of feature generators $g_s$ and $g_t$) such that $g\#P_s$ and $g\#P_t$ are close. Methods based on minimizing various distances between the two distributions (*e.g.*, maximum mean discrepancy [17, 44], suitable divergences and their approximations [15, 4, 35]) or matching the moments of the two distributions [41, 40] have been proposed for unsupervised domain adaptation.

## 2.2 Co-regularized Domain Alignment

The idea of co-regularization has been successfully used in semi-supervised learning [37, 38, 31, 36] for reducing the size of the hypothesis class. It works by learning two predictors in two hypothesis classes $\mathcal{H}_1$ and $\mathcal{H}_2$ respectively, while penalizing the disagreement between their predictions on the unlabeled examples. This intuitively results in shrinking the search space by ruling out predictors from $\mathcal{H}_1$ that don't have an agreeing predictor in $\mathcal{H}_2$ (and vice versa) [36]. When $\mathcal{H}_1$ and $\mathcal{H}_2$ are reproducing kernel Hilbert spaces, the co-regularized hypothesis class has been formally shown to have a reduced Rademacher complexity, by an amount that depends on a certain data dependent distance between the two views [31]. This results in improved generalization bounds comparing with the best predictor in the co-regularized class (reduces the variance) [2].

Suppose the true labeling functions for source and target domains are given by $f_s : X \to Y$ and $f_t : X \to Y$, respectively. Let $X_s^y = \{x : f_s(x) = y, x \sim P_s\}$ and $X_t^y = \{x : f_t(x) = y, x \sim P_t\}$

be the sets which are assigned label $y$ in source and target domains, respectively. As discussed in the earlier section, the hope is that alignment of marginal distributions $g\#P_s$ and $g\#P_t$ will result in aligning the corresponding class conditionals $g\#P_s(\cdot|y)$ and $g\#P_t(\cdot|y)$ but it is not guaranteed. There might be sets $A_s^{y_1} \subset X_s^{y_1}$ and $A_t^{y_2} \subset X_t^{y_2}$, for $y_1 \neq y_2$, such that their images under $g$ (*i.e.*, $g(A_s^{y_1}) := \{g(x) : x \in A_s^{y_1}\}$ and $g(A_t^{y_2}) := \{g(x) : x \in A_t^{y_2}\}$) get aligned in the feature space, which is difficult to detect or correct in the absence of target labels.

We propose to use the idea of co-regularization to trim the space of possible alignments without ruling out the desirable alignments of class conditional distributions from the space. Let $\mathcal{G}_1$, $\mathcal{G}_2$ be the two hypothesis spaces for the feature generators, and $\mathcal{H}_1$, $\mathcal{H}_2$ be the hypothesis classes of predictors defined on the output of the feature generators from $\mathcal{G}_1$ and $\mathcal{G}_2$, respectively. We want to learn a $g_i \in \mathcal{G}_i$ and a $h_i \in \mathcal{H}_i$ such that $h_i \circ g_i$ minimizes the prediction error on the source domain, while aligning the source and target feature distributions by minimizing a suitable distance $D(g_i\#P_s, g_i\#P_t)$ (for $i = 1, 2$). To measure the disagreement between the alignments of feature distributions in the two feature spaces ($g_i\#P_s$ and $g_i\#P_t$, for $i = 1, 2$), we look at the distance between the predictions $(h_1 \circ g_1)(x)$ and $(h_2 \circ g_2)(x)$ on unlabeled target examples $x \sim P_t$. If the predictions agree, it can be seen as an indicator that the alignment of source and target feature distributions is similar across the two feature spaces induced by $g_1$ and $g_2$ (with respect to the classifier boundaries). Coming back to the example of erroneous alignment given in the previous paragraph, if there is a $g_1 \in \mathcal{G}_1$ which aligns $g_1(A_t^{y_2})$ and $g_1(A_s^{y_1})$ but does not have any agreeing $g_2 \in \mathcal{G}_2$ with respect to the classifier predictions, it will be ruled out of consideration. Hence, ideally we would like to construct $\mathcal{G}_1$ and $\mathcal{G}_2$ such that they induce complementary erroneous alignments of source and target distributions while each of them still contains the set of desirable feature generators that produce the right alignments.

The proposed co-regularized domain alignment (referred as Co-DA) can be summarized by the following objective function (denoting $f_i = h_i \circ g_i$ for $i = 1, 2$):

$$\min_{\substack{g_i \in \mathcal{G}_i, h_i \in \mathcal{H}_i \\ f_i = h_i \circ g_i}} L_y(f_1; P_s) + \lambda_d L_d(g_1\#P_s, g_1\#P_t) + L_y(f_2; P_s) + \lambda_d L_d(g_2\#P_s, g_2\#P_t)$$

$$+ \lambda_p L_p(f_1, f_2; P_t) - \lambda_{div} D_g(g_1, g_2), \tag{2}$$

where, $L_y(f_i; P_s) := \mathbb{E}_{x,y\sim P_s}[y^\top \ln f_i(x)]$ is the usual cross-entropy loss for the source examples (assuming $f_i$ outputs the probabilities of classes and $y$ is the label vector), $L_d(\cdot, \cdot)$ is the loss term measuring the distance between the two distributions, $L_p(f_1, f_2; P_t) := \mathbb{E}_{x\sim P_t} l_p(f_1(x), f_2(x))$ where $l_p(\cdot, \cdot)$ measures the disagreement between the two predictions for a target sample, and $D_g(g_1, g_2)$ quantifies the diversity of $g_1$ and $g_2$. In the following, we instantiate Co-DA algorithmically, getting to a concrete objective that can be optimized.

### 2.2.1 Algorithmic Instantiation

We make our approach of co-regularized domain alignment more concrete by making the following algorithmic choices:

**Domain alignment.** Following much of the earlier work, we minimize the variational form of the Jensen-Shannon (JS) divergence [29, 18] between source and target feature distributions [15, 4, 35]:

$$L_d(g_i\#P_s, g_i\#P_t) := \sup_{d_i} \underbrace{\mathbb{E}_{x\sim P_s} \ln d_i(g_i(x)) + \mathbb{E}_{x\sim P_t} \ln(1 - d_i(g_i(x)))}_{L_{disc}(g_i, d_i; P_s, P_t)}, \tag{3}$$

where $d_i$ is the domain discriminator, taken to be a two layer neural network that outputs the probability of the input sample belonging to the source domain.

**Target prediction agreement.** We use $\ell_1$ distance between the predicted class probabilities (twice the total variation distance) as the measure of disagreement (although other measures such as JS-divergence are also possible):

$$L_p(f_1, f_2; P_t) := \mathbb{E}_{x\sim P_t} \|f_1(x) - f_2(x)\|_1 \tag{4}$$

**Diverse $g_1$ and $g_2$.** It is desirable to have $g_1$ and $g_2$ such that errors in the distribution alignments are different from each other and target prediction agreement can play its role. To this end, we encourage *source feature distributions* induced by $g_1$ and $g_2$ to be different from each other. There can be

multiple ways to approach this; here we adopt a simpler option of pushing the minibatch means (with batch size $b$) far apart:

$$D_g(g_1, g_2) := \min \left( \nu, \left\| \frac{1}{b} \sum_{j=1,\, x_j \sim P_s}^{b} (g_1(x_j) - g_2(x_j)) \right\|_2^2 \right) \tag{5}$$

The hyperparameter $\nu$ is a positive real controlling the maximum disparity between $g_1$ and $g_2$. This is needed for stability of feature maps $g_1$ and $g_2$ during training: we empirically observed that having $\nu$ as infinity results in their continued divergence from each other, harming the alignment of source and target distributions in both $\mathcal{G}_1$ and $\mathcal{G}_2$. Note that we only encourage the source feature distributions $g_1 \# P_s$ and $g_2 \# P_s$ to be different, hoping that aligning the corresponding target distributions $g_1 \# P_t$ and $g_2 \# P_t$ to them will produce different alignments.

**Cluster assumption.** The large amount of target unlabeled data can be used to bias the classifier boundaries to pass through the regions containing low density of data points. This is referred as the *cluster assumption* [7] which has been used for semi-supervised learning [19, 27] and was also recently used for unsupervised domain adaptation [35]. Minimization of the conditional entropy of $f_i(x)$ can be used to push the predictor boundaries away from the high density regions [19, 27, 35]. However, this alone may result in overfitting to the unlabeled examples if the classifier has high capacity. To avoid this, virtual adversarial training (VAT) [27] has been successfully used in conjunction with conditional entropy minimization to smooth the classifier surface around the unlabeled points [27, 35]. We follow this line of work and add the following additional loss terms for conditional entropy minimization and VAT to the objective in (2):

$$L_{ce}(f_i; P_t) := -\mathbb{E}_{x \sim P_t}[f_i(x)^\top \ln f_i(x)], \ L_{vt}(f_i; P_t) := \mathbb{E}_{x \sim P_t} \left[ \max_{\|r\| \leq \epsilon} D_{kl}(f_i(x) \| f_i(x + r)) \right] \tag{6}$$

We also use VAT loss $L_{vt}(f_i; P_s)$ on the source domain examples following Shu et al. [35]. Our final objective is given as:

$$\min_{g_i, h_i, f_i = h_i \circ g_i} \mathcal{L}(f_1) + \mathcal{L}(f_2) + \lambda_p L_p(f_1, f_2; P_t) - \lambda_{div} D_g(g_1, g_2), \ \text{where} \tag{7}$$

$$\mathcal{L}(f_i) := L_y(f_i; P_s) + \lambda_d L_d(g_i \# P_s, g_i \# P_t) + \lambda_{sv} L_{vt}(f_i; P_s) + \lambda_{ce}(L_{ce}(f_i; P_t) + L_{vt}(f_i; P_t))$$

**Remarks.**
**(1)** The proposed co-regularized domain alignment (Co-DA) can be used to improve any domain adaptation method that has a domain alignment component in it. We instantiate it in the context of a recently proposed method VADA [35], which has the same objective as $\mathcal{L}(f_i)$ in Eq. (7) and has shown state-of-the-art results on several datasets. Indeed, we observe that co-regularized domain alignment significantly improves upon these results.
**(2)** The proposed method can be naturally extended to more than two hypotheses, however we limit ourselves to two hypothesis classes in the empirical evaluations.

## 3 Related Work

**Domain Adaptation.** Due to the significance of domain adaptation in reducing the need for labeled data, there has been extensive activity on it during past several years. Domain alignment has almost become a representative approach for domain adaptation, acting as a crucial component in many recently proposed methods [17, 26, 41, 40, 4, 16, 44, 42, 35]. The proposed co-regularized domain alignment framework is applicable in all such methods that utilize domain alignment as an ingredient. Perhaps most related to our proposed method is a recent work by Saito et al. [33], who proposed directly optimizing a proxy for $\mathcal{H}\Delta\mathcal{H}$-distance [2] in the context of deep neural networks. Their model consists of a single feature generator $g$ that feeds to two different multi-layer NN classifiers $h_1$ and $h_2$. Their approach alternates between two steps: (i) For a fixed $g$, finding $h_1$ and $h_2$ such that the discrepancy or disagreement between the predictions $(h_1 \circ g)(x)$ and $(h_2 \circ g)(x)$ is maximized for $x \sim P_t$, (ii) For fixed $h_1$ and $h_2$, find $g$ which minimizes the discrepancy between the predictions $(h_1 \circ g)(x)$ and $(h_2 \circ g)(x)$ for $x \sim P_t$. Our approach also has a discrepancy minimization term over the predictions for target samples but the core idea in our approach is fundamentally different where we want to have diverse feature generators $g_1$ and $g_2$ that induce different alignments for source and target populations, and which can correct each other's errors by minimizing disagreement between them as measured by target predictions. Further, unlike [33] where the discrepancy is maximized at

the final predictions $(h_1 \circ g)(x)$ and $(h_2 \circ g)(x)$ (Step (i)), we maximize diversity at the output of feature generators $g_1$ and $g_2$. Apart from the aforementioned approaches, methods based on image translations across domains have also been proposed for unsupervised domain adaptation [24, 28, 6].

**Co-regularization and Co-training.** The related ideas of co-training [3] and co-regularization [37, 36] have been successfully used for semi-supervised learning as well as unsupervised learning [21, 20]. Chen et al. [8] used the idea of co-training for semi-supervised domain adaptation (assuming a few target labeled examples are available) by finding a suitable split of the features into two sets based on the notion of $\epsilon$-expandibility [1]. A related work [9] used the idea of co-regularization for semi-supervised domain adaptation but their approach is quite different from our method where they learn different classifiers for source and target, making their predictions agree on the unlabeled target samples. Tri-training [45] can be regarded as an extension of co-training [3] and uses the output of three different classifiers to assign pseudo-labels to unlabeled examples. Saito et al. [32] proposed Asymmetric tri-training for unsupervised domain adaptation where one of the three models is learned only on pseudo-labeled target examples. Asymmetric tri-training, similar to [33], works with a single feature generator $g$ which feeds to three different classifiers $h_1$, $h_2$ and $h_3$.

**Ensemble learning.** There is an extensive line of work on ensemble methods for neural nets which combine predictions from multiple models [11, 10, 30, 25, 23]. Several ensemble methods also encourage diversity among the classifiers in the ensemble [25, 23]. However, ensemble methods have a different motivation from co-regularization/co-training: in the latter, diversity and agreement go hand in hand, working together towards reducing the size of the hypothesis space and the two classifiers converge to a similar performance after the completion of training due to the agreement objective. Indeed, we observe this in our experiments as well and either of the two classifiers can be used for test time predictions. On the other hand, ensemble methods need to combine predictions from all member models to get desired accuracy which can be both memory and computation intensive.

# 4 Experiments

We evaluate the proposed Co-regularized Domain Alignment (Co-DA) by instantiating it in the context of a recently proposed method VADA [35] which has shown state-of-the-art results on several benchmarks, and observe that Co-DA yields further significant improvement over it, establishing new state-of-the-art in several cases. For a fair comparison, we evaluate on the same datasets as used in [35] (*i.e.*, MNIST, SVHN, MNIST-M, Synthetic Digits, CIFAR-10 and STL), and base our implementation on the code released by the authors[3] to rule out incidental differences due to implementation specific details.

**Network architecture.** VADA [35] has three components in the model architecture: a feature generator $g$, a feature classifier $h$ that takes output of $g$ as input, and a domain discriminator $d$ for domain alignment (Eq. 3). Their data classifier $f = h \circ g$ consists of nine `conv` layers followed by a global pool and `fc`, with some additional dropout, max-pool and Gaussian noise layers in $g$. The last few layers of this network (the last three `conv` layers, global pool and `fc` layer) are taken as the feature classifier $h$ and the remaining earlier layers are taken as the feature generator $g$. Each `conv` and `fc` layer in $g$ and $h$ is followed by batch-norm. The objective of VADA for learning a data classifier $f_i = h_i \circ g_i$ is given in Eq. (7) as $\mathcal{L}(f_i)$. We experiment with the following two architectural versions for creating the hypotheses $f_1$ and $f_2$ in our method: **(i)** We use two VADA models as our two hypotheses, with each of these following the same architecture as used in [35] (for all three components $g_i$, $h_i$ and $d_i$) but initialized with different random seeds. This version is referred as **Co-DA** in the result tables. **(ii)** We use a single (shared) set of parameters for `conv` and `fc` layers in $g_1/g_2$ and $h_1/h_2$ but use conditional batch-normalization [12] to create two different sets of batch-norm layers for the two hypotheses. However we still have two different discriminators (unshared parameters) performing domain alignment for features induced by $g_1$ and $g_2$. This version is referred as **Co-DA**$^{bn}$ in the result tables. Additionally, we also experiment with fully shared networks parameters without conditional batch-normalization (*i.e.*, shared batchnorm layers): in this case, $g_1$ and $g_2$ differ only due to random sampling in each forward pass through the model (by virtue of the dropout and Gaussian noise layers in the feature generator). We refer this variant as **Co-DA**$^{sh}$ (for **sh**ared parameters). The diversity term $D_g$ (Eq. (5)) becomes inapplicable in this case. This also

has resemblance to $\Pi$-model [22] and *fraternal dropout* [46], which were recently proposed in the context of (semi-)supervised learning.

**Other details and hyperparameters.** For domain alignment, which involves solving a saddle point problem ($\min_{g_i} \max_{d_i} L_{disc}(g_i, d_i; P_s, P_t)$, as defined in Eq. 3), Shu et al. [35] replace gradient reversal [15] with alternating minimization ($\max_{d_i} L_{disc}(g_i, d_i; P_s, P_t)$, $\min_{g_i} L_{disc}(g_i, d_i; P_s, P_t)$) as used by Goodfellow et al. [18] in the context of GAN training. This is claimed to alleviate the problem of saturating gradients, and we also use this approach following [35]. We also use *instance normalization* following [35] which helps in making the classifier invariant to channel-wide shifts and scaling of the input pixel intensities. We do not use any sort of data augmentation in any of our experiments. For VADA hyperparameters $\lambda_{ce}$ and $\lambda_{sv}$ (Eq. 7), we fix their values to what were reported by Shu et al. [35] for all the datasets (obtained after a hyperparameter search in [35]). For the domain alignment hyperparameter $\lambda_d$, we do our own search over the grid $\{10^{-3}, 10^{-2}\}$ (the grid for $\lambda_d$ was taken to be $\{0, 10^{-2}\}$ in [35]). The hyperparameter for target prediction agreement, $\lambda_p$, was obtained by a search over $\{10^{-3}, 10^{-2}, 10^{-1}\}$. For hyperparameters in the diversity term, we fix $\lambda_{div} = 10^{-2}$ and do a grid search for $\nu$ (Eq. 5) over $\{1, 5, 10, 100\}$. The hyperparameters are tuned by randomly selecting 1000 target labeled examples from the training set and using that for validation, following [35, 32]. We completely follow [35] for training our model, using Adam Optimizer ($lr = 0.001$, $\beta_1 = 0.5$, $\beta_2 = 0.999$) with Polyak averaging (*i.e.*, an exponential moving average with momentum$= 0.998$ on the parameter trajectory), and train the models in all experiments for $80k$ iterations as in [35].

**Baselines.** We primarily compare with VADA [35] to show that co-regularized domain alignment can provide further improvements over state-of-the-art results. We also show results for Co-DA without the diversity loss term (*i.e.*, $\lambda_{div} = 0$) to tease apart the effect of explicitly encouraging diversity through Eq. 5 (note that some diversity can arise even with $\lambda_{div} = 0$, due to different random seeds, and Gaussian noise / dropout layers present in $g$). Shu et al. [35] also propose to incrementally refine the learned VADA model by shifting the classifier boundaries to pass through low density regions of target domain (referred as the DIRT-T phase) while keeping it from moving too far away. If $f^{*n}$ is the classifier at iteration $n$ ($f^{*0}$ being the solution of VADA), the new classifier is obtained as $f^{*n+1} = \arg\min_{f^{n+1}} \lambda_{ce}(L_{ce}(f^{n+1}; P_t) + L_{vt}(f^{n+1}; P_t)) + \beta \mathbb{E}_{x \sim P_t} D_{kl}(f^{*n}(x) \| f^{n+1}(x))$. We also perform DIRT-T refinement individually on each of the two trained hypotheses obtained with Co-DA (*i.e.*, $f_1^{*0}, f_2^{*0}$) to see how it compares with DIRT-T refinement on the VADA model [35]. Note that DIRT-T refinement phase is carried out individually for $f_1^{*0}$ and $f_2^{*0}$ and there is no co-regularization term connecting the two in DIRT-T phase. Again, following the evaluation protocol in [35], we train DIRT-T for $\{20k, 40k, 60k, 80k\}$ iterations, with number of iterations taken as a hyperparameter. We do not perform any hyperparameter search for $\beta$ and the values for $\beta$ are fixed to what were reported in [35] for all datasets. Apart from VADA, we also show comparisons with other recently proposed unsupervised domain adaptation methods for completeness.

## 4.1 Domain adaptation results

We evaluate Co-DA on the following domain adaptation benchmarks. The results are shown in Table 1. The two numbers A/B in the table for the proposed methods are the individual test accuracies for both classifiers which are quite close to each other at convergence.

**MNIST$\rightarrow$SVHN.** Both MNIST and SVHN are digits datasets but differ greatly in style : MNIST consists of gray-scale handwritten digits whereas SVHN consists of house numbers from street view images. This is the most challenging domain adaptation setting in our experiments (many earlier domain adaptation methods have omitted it from the experiments due to the difficulty of adaptation). VADA [35] showed good performance (73.3%) on this challenging setting using instance normalization but without using any data augmentation. The proposed Co-DA improves it substantially $\sim 81.7\%$, even surpassing the performance of VADA+DIRT-T (76.5%) [35]. Figure 2 shows the test accuracy as training proceeds. For the case of no instance-normalization as well, we see a substantial improvement over VADA from $47.5\%$ to $52\%$ using Co-DA and $55.3\%$ using Co-DA$^{bn}$. Applying iterative refinement with DIRT-T [35] further improves the accuracy to $88\%$ with instance norm and $\sim 60\%$ without instance norm. This sets new state-of-the-art for MNIST$\rightarrow$SVHN domain adaptation without using any data augmentation. To directly measure the improvement in source and target feature distribution alignment, we also do the following experiment: (i) We take the feature embeddings $g_1(\cdot)$ for the source training examples, reduce the dimensionality to $50$ using PCA, and

| Source<br>Target | MNIST<br>SVHN | SVHN<br>MNIST | MNIST<br>MNIST-M | DIGITS<br>SVHN | CIFAR<br>STL | STL<br>CIFAR |
|---|---|---|---|---|---|---|
| DANN [15] | 35.7 | 71.1 | 81.5 | 90.3 | - | - |
| DSN [4] | - | 82.7 | 83.2 | 91.2 | - | - |
| ATT [32] | 52.8 | 86.2 | 94.2 | 92.9 | - | - |
| MCD [33] | - | 96.2 | - | - | - | - |
| *Without instance-normalized input* | | | | | | |
| VADA [35] | 47.5 | 97.9 | 97.7 | 94.8 | 80.0 | 73.5 |
| Co-DA ($\lambda_{div} = 0$) | 50.7/50.1 | 97.4/97.2 | 98.9/99.0 | 94.9/94.6 | 81.3/80 | 76.1/75.5 |
| Co-DA$^{bn}$ ($\lambda_{div} = 0$) | 46.0/45.9 | 98.4/98.3 | **99.0/99.0** | 94.9/94.8 | 80.4/80.3 | 76.3/76.6 |
| Co-DA$^{sh}$ | 52.8 | *98.6* | 98.9 | **96.1** | 78.9 | 76.1 |
| Co-DA | 52.0/49.7 | 98.3/98.2 | *99.0/98.9* | *96.1/96.0* | *81.1/80.4* | **76.4/75.7** |
| Co-DA$^{bn}$ | **55.3/55.2** | **98.8/98.7** | 98.6/98.7 | 95.4/95.3 | **81.4/81.2** | *76.3/76.2* |
| VADA+DIRT-T [35] | 54.5 | *99.4* | 98.9 | 96.1 | - | 75.3 |
| Co-DA+DIRT-T | 59.8/60.8 | **99.4/99.4** | **99.1/99.0** | **96.4/96.5** | - | 76.3/76.6 |
| Co-DA$^{bn}$+DIRT-T | **62.4/63.0** | 99.3/99.2 | *98.9/99.0* | 96.1/96.1 | - | **77.6/77.5** |
| *With instance-normalized input* | | | | | | |
| VADA [35] | 73.3 | 94.5 | 95.7 | 94.9 | 78.3 | 71.4 |
| Co-DA ($\lambda_{div} = 0$) | 78.5/78.2 | 97.6/97.5 | 97.1/96.4 | 95.1/94.9 | 80.1/79.2 | 74.5/73.9 |
| Co-DA$^{bn}$ ($\lambda_{div} = 0$) | 74.5/74.3 | 98.4/98.4 | 96.7/96.6 | 95.3/95.2 | 78.9/79.0 | 74.2/74.4 |
| Co-DA$^{sh}$ | 79.9 | **98.7** | 96.9 | **96.0** | 78.4 | **74.7** |
| Co-DA | **81.7/80.9** | *98.6/98.5* | 97.5/97.0 | *96.0/95.9* | 80.6/79.9 | 74.7/74.2 |
| Co-DA$^{bn}$ | *81.4/81.3* | 98.5/98.5 | **98.0/97.9** | 95.3/95.3 | **80.6/80.4** | *74.7/74.6* |
| VADA+DIRT-T [35] | 76.5 | 99.4 | 98.7 | 96.2 | - | 73.3 |
| Co-DA+DIRT-T | **88.0/87.3** | 99.3/99.4 | 98.7/98.6 | 96.4/96.5 | - | 74.8/74.2 |
| Co-DA$^{bn}$+DIRT-T | 86.5/86.7 | 99.4/99.3 | 98.8/98.8 | 96.4/96.5 | - | **75.9/75.6** |

Table 1: Test accuracy on the Target domain: Co-DA$^{bn}$ is the proposed method for the two classifiers with shared parameters but with different batch-norm layers and different domain discriminators. Co-DA$^{sh}$ is another variant where the only sources of difference between the two classifiers are the stochastic layers (dropout and Gaussian noise). The stochastic layers collapse to their expectations and we effectively have a single classifier during test phase. For Co-DA, the two numbers A/B are the accuracies for the two classifiers (at $80k$ iterations). Numbers in **bold** denote the best accuracy among the comparable methods and those in *italics* denote the close runner-up, if any. VADA and DIRT-T results are taken from [35].

use these as training set for a k-nearest neighbor (kNN) classifier. (ii) We then compute the accuracy of this kNN classifier on target test sets (again with PCA on the output of feature generator).We also do steps (i) and (ii) for VADA, and repeat for multiple values of 'k'. Fig. 3 compares the target test accuracy scores for VADA and Co-DA.

**SVHN→MNIST.** This adaptation direction is easier as MNIST as the test domain is easy to classify and performance of existing methods is already quite high ($97.9\%$ with VADA). Co-DA is still able to yield reasonable improvement over VADA, of about $\sim 1\%$ for no instance-normalization, and $\sim 4\%$ with instance-normalization. The application of DIRT-T after Co-DA does not give significant boost over VADA+DIRT-T as the performance is already saturated with Co-DA (close to $99\%$).

**MNIST→MNIST-M.** Images in MNIST-M are created by blending MNIST digits with random color patches from the BSDS500 dataset. Co-DA provides similar improvements over VADA as the earlier setting of SVHN→MNIST, of about $\sim 1\%$ for no instance-normalization, and $\sim 2\%$ with instance-normalization.

**Syn-DIGITS→SVHN.** Syn-DIGITS data consists of about $50k$ synthetics digits images of varying positioning, orientation, background, stroke color and amount of blur. We again observe reasonable improvement of $\sim 1\%$ with Co-DA over VADA, getting close to the accuracy of a fully supervised target model for SVHN (without data augmentation).

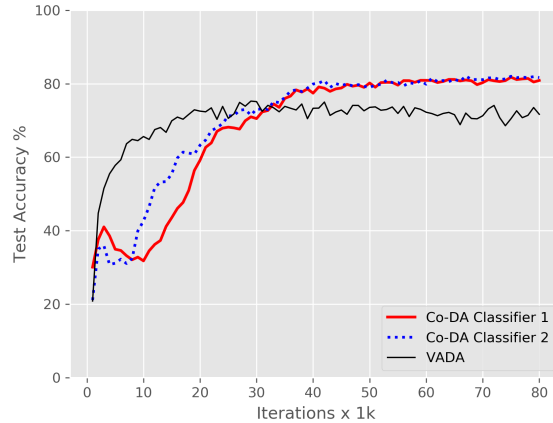

Figure 2: Test accuracy as the training iterations proceed for MNIST→SVHN with instance-normalization: there is high disagreement between the two classifiers during the earlier iterations for Co-DA, which vanishes eventually at convergence. VADA [35] gets to a much higher accuracy early on during training but eventually falls short of Co-DA performance.

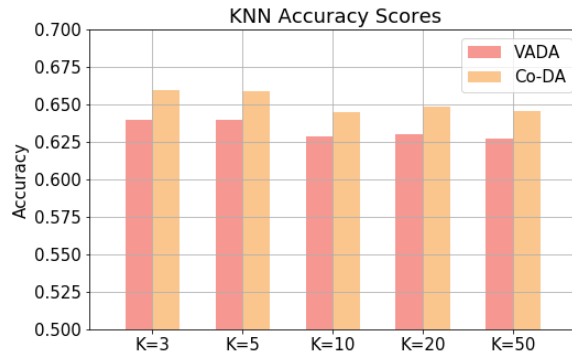

Figure 3: Test accuracy of a kNN classifier on target domain for VADA and Co-DA: source domain features (output of $g_1(\cdot)$, followed by PCA which reduces dimensionality to $50$) are used as training data for the classifier.

**CIFAR↔STL.** CIFAR has more labeled examples than STL hence CIFAR→STL is easier adaptation problem than STL→CIFAR. We observe more significant gains on the harder problem of STL→CIFAR, with Co-DA improving over VADA by $3\%$ for both with- and without instance-normalization.

## 5 Conclusion

We proposed co-regularization based domain alignment for unsupervised domain adaptation. We instantiated it in the context of a state-of-the-art domain adaptation method and observe that it provides improved performance on some commonly used domain adaptation benchmarks, with substantial gains in the more challenging tasks, setting new state-of-the-art in these cases. Further investigation is needed into more effective diversity losses (Eq. (5)). A theoretical understanding of co-regularization for domain adaptation in the context of deep neural networks, particularly characterizing its effect on the alignment of source and target feature distributions, is also an interesting direction for future work.

## Footnotes

[1]Heavily-tuned manual data augmentation can be used to bring the two domains closer in the observed space $X$ [14] but it requires the augmentation to be tuned individually for every domain pair to be successful.

[2]Sridharan and Kakade [38] show that the bias introduced by co-regularization is small when each view carries sufficient information about $Y$ on its own (*i.e.*, mutual information $I(Y; X_1|X_2)$ and $I(Y; X_2|X_1)$ are small), and the generalization bounds comparing with the Bayes optimal predictor are also tight.

[3]`https://github.com/RuiShu/dirt-t`

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
