[Reviews · NeurIPS 2018]

Reviewer 1



This paper presents a co regularization approach to domain adaptation. They build upon the "VADA" framework by A dirt-t approach to unsupervised domain adaptation. In particular, the VADA framework trains feature extractor 'g' and classifier 'h' by minimizing: distributional distance between features extracted from the source and target domains, label loss on source domain, and entropy + a smoothness operator over the output function (g*h) on the source and target domains. The contribution of this paper is to first point out that alignment between source and target distributions is possible even with a 'g' that does not align the class distributions across source and target, and vice versa. To this end, the paper proposes to use co-regularization: learn two feature extractors g1 and g2 and two classifiers h1 and h2, via the VADA objective function, while also minimizing the distance between g1*h1 and g2*h2 and maximizing the distance between g1 and g2 for diversity. The authors claim that this can help rule out incorrect alignments. Cons: 1. This paper argues in section 1 and figure 1 how alignment is not guaranteed to reduce target error and vice versa. However, with their co-regularization framework, they do not rigorously show (either theoretically or with controlled empirical results) that their approach indeed results in better alignment. The arguments on lines 112-116 and 123-128 are quite hand-wavy. This is indeed my biggest gripe with the paper because it is not quite clear how is the co-regularization actually helping in this case. In the co-regularization paper by Rosenberg and Bartlett, the use of two separate RKHS was shown to reduce Rademacher complexity, but what exactly is going on in this case? 2. More of a comment on above than a con: The experimental results show that even with \lambda_div=0, their framework provides improvements. The diversity in g1 and g2, in this case, is only coming from randomness (as the authors themselves point out on lines 242-245). In effect, this relies on the randomness of initialization and training along with non-convexity of the objective function to generate diversity. This needs to be explained and investigated further -- can we just rule out more and more "incorrect alignments" by randomly generating classifiers. If so, can we just make the feature generator better by learning 3 instead of 2 classifiers? Question; 1. What set of parameters are finally used for testing on the target layer? After author feedback: Upgraded my score to 6 provided the authors add their kNN explanation to the paper.

Reviewer 2



This paper addresses the topic of unsupervised domain adaptation, i.e. how to adapt a classifier which was trained for one task to another one using unlabeled data of the target task only. There is a large body of research on this topic. Many algorithms are based on the alignment of the source and target feature distributions. This paper proposes to extend this idea by constructing multiple feature spaces and aligning them individually, while encouraging agreement between them. This approach is surprisingly successful and outperforms the state-of-the-art on the digit classification tasks MNIST, MINSTM, SVHN, DIGITS, STL and CIFAR. In Table 1, you provide the results for both classifiers. How do you select the one that you would use in a real application ? One could probably chose either one since their performance are always very close. Can this be guaranteed ? Since we have two classifiers, one also wonders whether we could get additional improvements by combining them, e.g. by majority vote or averaging the predictions. You mention that you only use two classifiers in this paper. Obviously, the question arises what would happen when you used more. Do you expect additional improvements ? Or is this computationally intractable ? How would you align multiple feature spaces - pairwise or with respect to one "master feature space". It would be nice to comment on these topics in the next version of this paper, with our without running experiments.

Reviewer 3



Title: Co-regularized Alignment for Unsupervised Domain Adaptation Summary Unsupervised domain adaptation aims at matching the feature distribution of a a collection of labeled data from a source domain with unlabeled examples from a target domain to enhance generalization to unseen data. However, matching the marginal feature distribution may not entail that the corresponding class distribution will be aligned across the source and target domain as well. To address this issue, the authors suggest to construct multiple diverse feature spaces where the alignment between source and target feature distributions is aligned for each space individually while also encouraging that alignments to agree with each other according to the labels of the examples of the source domain and the class predictions of the unlabeled examples from the target domain. The suggested method improves upon several domain adaptation benchmarks Strengths Method advances state-of-the-art on several benchmarks. CoDA is fairly intuitive and easy to understand. Excellent survey of related work. Implementation and baselines are based on a publicly available code base. Weaknesses Experiments do not explore the construction of more than two feature spaces. With the exception of SVHN, the utilized benchmarks are fairly simple classification tasks. However, this is a common practice in the unsupervised domain adaptation community and not a particular issue with this work. No commitment expressed to release a reference implementation of this method. Questions Comments Editorial Notes L212: later -> latter ------------------------- Thanks to the authors for their detailed responses. The responses reaffirmed my assessment and I will stay with my original overall score.

Reviewer 4



Given a domain adaptation algorithm that aligns the source and target feature spaces, the paper suggests leveraging the adaptation by learning several distinct alignments while enjoining them to agree on unlabeled target samples. It implements this idea by adding new losses to already successful domain adaptation neural network models. The idea is appealing and the empirical results are convincing. I appreciate that the experiments reuse the recent network architecture and training procedure of Shu et al. (ICML 2018). It allows to truly isolate the benefit of the new method. It should be easy to reuse it in many contexts by adding to existing neural networks the new components. For these reasons, I tend to suggest paper acceptance. However, I think the contribution could be greater, and does not provide enough insights (either theoretical or empirical) to understand in which context the method is working. Section 2.1 discusses the domain adaptation bound of Ben-David et al. (2006) (as many previous domain adaptation papers, but the proposed co-alignment algorithm is not really supported by the cited theorem (at least it is not discussed in the paper). The novelties of the paper are presented very succinctly in less than a half page: The target prediction alignment of Equation 4 and the diversity component of Equation 5. In both cases, the authors mention that many strategies could be used to implement these concepts. Their choices should be discussed more. In particular, few is said about the diversity component of Equation 5 (the equation itself is ambiguous, as x_j is drawn according to P_s and j iterates from 1 to b. Also, b is undefined). What is the influence of the minibatch size? How can we extend this equation to more than two alignments? On the theoretical side, the role of the disagreement on the unlabeled target sample could be related to the analysis of the following paper: "A New PAC-Bayesian Perspective on Domain Adaptation" (Germain et al., 2016). That being said, the best way to give insights about the proposed method would probably be to provide a well-designed toy example where we can see the benefit of co-alignment in action. Minor comments: * The fact that the objective function is presented twice is confusing. It is first said that CoDA is summarized by Equation 2, and we have to continue to read to learn that it is in fact given by Equation 7 (including new losses) * Please choose between "CoDA" or "Co-DA" * References [14] and [15] seems to refer to the same paper